# Regulating TiO_2_ Deposition Using a Single-Anchored Ligand for High-Efficiency Perovskite Solar Cells

**DOI:** 10.3390/ma17153820

**Published:** 2024-08-02

**Authors:** Zhanpeng Xu, Zhineng Lan, Fuxin Chen, Chong Yin, Longze Wang, Zhehan Li, Luyao Yan, Jun Ji

**Affiliations:** 1Power China Huadong Engineering Corporation Limited, Hangzhou 311122, China; 2State Key Laboratory of Alternate Electrical Power System with Renewable Energy Sources, North China Electric Power University, Beijing 102206, China; 3Beijing Huairou Laboratory, Beijing 101400, China

**Keywords:** perovskite solar cells, electron transport layer, interfacial electron transport, TiO_2_

## Abstract

Planar perovskite solar cells (PSCs), as a promising photovoltaic technology, have been extensively studied, with strong expectations for commercialization. Improving the power conversion efficiency (PCE) of PSCs is necessary to accelerate their practical application, in which the electron transport layer (ETL) plays a key part. Herein, a single-anchored ligand of phenylphosphonic acid (PPA) is utilized to regulate the chemical bath deposition of a TiO_2_ ETL, further improving the PCE of planar PSCs. The PPA possesses a steric benzene ring and a phosphoric acid group, which can inhibit the particle aggregation of the TiO_2_ film through steric hindrance, leading to optimized interface (ETL/perovskite) contact. In addition, the incorporated PPA can induce the upshift of the Fermi-level of the TiO_2_ film, which is beneficial for interfacial electron transport. As a consequence, the PSCs with PPA-TiO_2_ achieve a PCE of 24.83%, which is higher than that (24.21%) of PSCs with TiO_2_. In addition, the unencapsulated PSCs with PPA-TiO_2_ also exhibit enhanced stability when stored in ambient conditions.

## 1. Introduction

Metal halide perovskite solar cells (PSCs) show huge potential in capturing and converting solar energy efficiently due to distinguishing advantages like low-cost solution preparation, high efficiency, and so on [1,2]. In 2009, PSCs were first reported with a power conversion efficiency (PCE) of 3.8% [3]. 

After massive efforts from the perspective of experimental and theory calculations on PSCs over the past years, researchers have achieved a substantial improvement in the performance of PSCs by developing a variety of optimization strategies, with the latest PCE certification rate of 26.14% [4]. However, compared to the Shockley–Queisser limit, there is still much room for PCE improvement. As a multilayered structure, the PCE is a comprehensive factor of PSCs, and it is related to all the function layers and their cooperation [5,6]. Among the function layers in n-i-p planar PSCs, the electron transport layer (ETL) plays a vital part in the PCE, since it not only extracts the photo-generated electron but also serves as a substrate for perovskite film deposition [7,8]. Hence, optimizing ETL properties and further exploring its underlying mechanism on PSC physical characteristics is supposed to be a feasible approach to improve the PCE of PSCs. 

Among the efficient planar PSCs, one of the popular ETLs is titanium dioxide (TiO_2_), which was applied to PSCs in the initial stage of PSC research [9,10,11,12]. In order to obtain a TiO_2_ ETL that can help achieve high-performance PSCs, researchers have developed preparation methods such as spin-coating, magnetron sputtering, and chemical bath deposition (CBD) [13,14,15]. Among them, CBD possesses the advantage of low-temperature deposition and conformal deposition, which is suitable for the light-managing textured substrate of FTO [16,17,18]. As early as 2019, Cui et al. [19] applied TiO_2_ prepared using CBD as an ETL, realizing a PCE of up to 21.88% in a planar PSC. In 2022, although combining the surface modification of TiO_2_ deposited using CBD and perovskite crystallization regulation, Li et al. [20] achieved a PCE of 24.5% but did not reach their limit value. Further breakthroughs can be achieved, especially considering that SnO_2_-based PSCs have achieved a certificated PCE of more than 25% in the same term. In fact, in the CBD process of TiO_2_, the hydrolysis activation energy of TiCl_4_ is low, and the reaction easily occurs. Accompanied by a large amount of exothermic and hydrochloric acid gas, the reaction process is very rapid, and a large number of TiO_2_ particles are generated and agglomerate in a short time, resulting in the formation of TiO_2_ films with a large number of oxygen vacancies with a rough morphology [21,22]. These vacancy defects and surface particle aggregation restrict PCE improvement through carrier recombination and inferior TiO_2_/perovskite interfacial contact properties [23]. Hence, it is necessary to regulate CBD and explore the functional mechanism of the regulation of TiO_2_ properties, realizing a high-quality TiO_2_ ETL to improve the photovoltaic performance of planar PSCs.

Herein, we regulated TiO_2_ CBD by incorporating phenylphosphonic acid (PPA; its molecular structure formula is shown in Appendix A) into a bath precursor mixed with TiCl_4_ and deionized water. The regulated TiO_2_ is named PPA-TiO_2_. The PPA possesses a steric benzene ring and a phosphoric acid group, which can interact with TiO_2_, achieving the inhibition of particle aggregation through steric hindrance. Additionally, the deposition regulation using PPA induces an upshift of the Fermi level of TiO_2_, optimizing interface energy level matching. After deposition regulation, the corresponding planar PSCs possess enhanced electron transport, which should result from the modified matching of the interfacial energy level and optimized interfacial contact property resulting from a smooth TiO_2_ surface. As a consequence, the PSCs with PPA-TiO_2_ achieve a champion PCE of 24.83%, which is higher than that (24.21%) of PSCs with TiO_2_. In addition, PSCs with PPA-TiO_2_ also exhibit enhanced storage stability.

## 2. Results and Discussion

CBD is a common method to deposit metal oxide films such as TiO_2_, tin dioxide (SnO_2_), and so on [24,25]. In this work, we utilized CBD to deposit a TiO_2_ film with the experimental parameters based on our previous reports [19]. First, we prepared the bath precursor by mixing the TiCl_4_ and deionized water. Then, we immersed the cleaned substrate into the precursor to grow the film. For regulating CBD, we introduced PPA as a representative example into the chemical bath precursor before the subsequent film growth. Appendix A shows the molecular structure formula of PPA, which has a phosphoric acid group that can be used as a ligand to coordinate with TiO_2_ to regulate the deposition process. First, we carried out X-ray photoelectron spectroscopy (XPS) measurements on the TiO_2_ and PPA-TiO_2_ films to obtain the surface elements and chemical states. As shown in Figure 1a, we can obviously observe the P 2p peaks near 132.5 eV on the PPA-TiO_2_ films, indicating the successful incorporation of PPA. Meanwhile, as shown in Figure 1b, the binding energies of the Ti 2p_1/2_ and Ti 2p_3/2_ peaks of the TiO_2_ ETL and PPA-TiO_2_ ETL are 464.09 and 458.27 eV and 464.19 and 458.37 eV, respectively. The shift of the Ti 2p peaks to higher binding energies indicates that PPA can form a chemical interaction with Ti. This interaction between PPA and TiO_2_ can be utilized to regulate the chemical bath process.

The influence of PPA on the TiO_2_ morphology was characterized using scanning electron microscope (SEM) and atomic force microscope (AFM) measurements. As shown in Figure 1c,d, the particle aggregation of TiO_2_ can be clearly observed, which was a rough topography. The poor surface morphology caused by the aggregation of the TiO_2_ particles would affect the electrical conductivity and interface contact between TiO_2_ and the perovskite film. In comparison, the PPA-TiO_2_ surface was relatively flat and smooth without obvious particle aggregation, which realizes conformal coverage on the FTO substrate. The AFM images of TiO_2_ and PPA-TiO_2_ are shown in Figure 1e,f. It can be found that there are some bright sites in the AFM image of TiO_2_, indicating existing particle aggregation, which is consistent with the SEM image (Figure 1c). As a comparison, the PPA-TiO_2_ was flatter with uniformly dispersed particles. Furthermore, the root mean square (RMS) of the surface topography height was calculated to analyze the surface roughness of the TiO2 and PPA-TiO2 films quantitatively. The RMS of PPA-TiO_2_ was 24 nm, which is obviously lower than that of TiO_2_ (34 nm), indicating the inhibition of particle aggregation on the film surface. The cross-sectional SEM images of the TiO_2_ film and PPA-TiO_2_ film are shown in Figure 1g,h. In Figure 1g, we can observe that the TiO_2_ film yields a rough surface, and due to the formation of agglomeration, it shows sharp ups and downs in different positions. In Figure 1h, it is observed that the PPA-TiO_2_ film maintains a relatively flat morphology in the entire section without particularly violent height fluctuations, and the entire layer of TiO_2_ appears very dense and smooth, which confirms its inhibitory effect on particle aggregation. The optical transmittance spectra of the TiO_2_ film and PPA-TiO_2_ film are shown in Appendix A, and the results show that the flatter surface also helps to reduce the scattering of incident light by the ETL, especially in the wavelength range of 320–380 nm, which helps to enhance the absorption of light by the perovskite light absorption layer.

Next, we investigated the influence of PPA on the electrical properties and energy level structure of the TiO_2_ films. Conductive atomic force microscopy (C-AFM) was used to evaluate the conductivity and the lateral uniformity. C-AFM uses a probe to conduct contact scanning with the sample surface and apply a bias voltage to generate Coulomb repulsion between atoms, obtain surface morphology and current intensity signals, and obtain nanoscale *J*–*V* curves of the film surface with an atomic-level resolution. It can be seen from Figure 2a,b that in addition to the significantly improved current, the PPA-TiO_2_ film obtained better uniformity in terms of current distribution. The *J*–*V* curves also reflect that the PPA-TiO_2_ film had optimized electrical conductivity (Figure 2c). Kelvin probe force microscopy (KPFM) measurements were used to detect the surface potentials of the films (Figure 2d,e). Compared with TiO_2_, the PPA-TiO_2_ film showed lower surface potential, suggesting a higher Fermi level (*E*_F_), which is favorable for interfacial electron transport. We also carried out an ultraviolet photoelectron spectroscopy (UPS) test to characterize the energy level structure of the TiO_2_ film and PPA-TiO_2_ film. Combining the results of the UPS and Tauc plots (Appendix A), we calculated the valence band maximum energy (*E*_VBM_) of the TiO_2_ film and PPA-TiO_2_ film to be −8.11 and −7.79 eV, respectively, and the conduction band minimum energy (*E*_CBM_) of the TiO_2_ film and PPA-TiO_2_ film to be −4.66 and −4.37 eV, respectively (Figure 2f). Compared to the TiO_2_ film, the *E*_F_ of the PPA-TiO_2_ film rose from −5.13 to −5.12 eV, and the *E*_CBM_ of the PPA-TiO_2_ film rose from −4.66 to −4.37 eV, which is conducive to optimizing interfacial electron transport and decreasing interfacial electron accumulation.

Since TiO_2_ is an ETL in n-i-p planar PSCs in which the ETL serves as the substrate for perovskite deposition, the surface topography of the TiO_2_ film has a significant effect on the crystallization growth process of the upper perovskite, thus affecting the quality of the perovskite film. To characterize the perovskite films fabricated on TiO_2_ and PPA-TiO_2_, we applied cross-sectional SEM, with the results shown in Figure 3a,b. Compared with the TiO_2_-based perovskite film, the perovskite film grown on PPA-TiO_2_ had a larger grain size with a uniform orientation, which should be attributed to the inhibited surface particle aggregation. In order to more directly observe the contact between the ETLs and perovskite and the growth of perovskite at the bottom, we performed SEM tests on the buried perovskite films by exposing the buried interface. The method of exposing the buried interface is shown in Figure 3c. We first used an epoxy adhesive to tighten the adhesion glass and the surface of the perovskite film. Due to the loose contact between the ETL and the buried perovskite, the buried interface of the perovskite film could be exposed through the holistic stripping of the glass and perovskite film. It can be clearly seen from Figure 3d,e that the buried interface of perovskite showed a smaller grain size based on the TiO_2_ ETL, while the perovskite film had a larger grain size based on PPA-TiO_2_, which is consistent with the cross-section SEM, indicating that PPA-TiO_2_ contributes to the crystal growth of perovskite. More importantly, there were a large number of TiO_2_ dendrites attached to the buried interface based on the TiO_2_ ETL, and there were more holes. In contrast, the buried interface based on the PPA-TiO_2_ ETL had fewer TiO_2_ dendrites and holes. This obvious difference can be attributed to the existence of a large number of loose TiO_2_ dendrite aggregation structures on the surface of the TiO_2_ ETL, and the use of PPA for deposition regulation could effectively control the deposition process of TiO_2_, reduce the generation of TiO_2_ dendrites, and achieve the tight deposition of TiO_2_ based on coordination with PPA.

Further, we carried out photoluminescence (PL) measurements to study the electron transport behavior at the interface between the ETL and perovskite. As shown in Figure 3f, the PL spectral intensity of the PPA-TiO_2_-based perovskite films decreased significantly, which indicates the enhanced electron transport from perovskite to the ETL. Mott–Schottky curves were also created to evaluate the physical properties of PSCs structured as fluorine-doped tin oxide (FTO)/TiO_2_ or PPA-TiO_2_/perovskite/methoxy-phenethylammonium iodide (MeO-PEAI)/2,2′,7,7′-tetrakis[N,N-di(4-methoxyphenyl)amino]-9-9′-spirobifluorene (Spiro-OMeTAD)/gold (Au). As shown in Figure 3g, the built-in electric field of the PPA-TiO_2_-based PSC was 1.06 V stronger than that of the TiO_2_-based PSC (1.04 V), which should have resulted from the optimized energy level structure of the TiO_2_ ETL. A stronger built-in electric field is beneficial for photo-generated carrier separation and transport to charge transport layers (ETLs and hole transport layers). In addition, as shown in Figure 3h, the dark *J*–*V* curves display that the dark saturation current of PSCs with PPA-TiO_2_ is smaller than the PSCs with TiO_2_, which indicates the reduced non-radiative recombination after regulating TiO_2_ deposition using PPA. This reduced non-radiative recombination should result from the decreased interfacial electron accumulation. The above results exhibit that the PPA-TiO_2_ can promote interfacial electron transport and reduce carrier non-radiative recombination, which is expected to show a positive effect on the photovoltaic performance of PSCs.

We fabricated planar PSCs structured as FTO/TiO_2_ (PPA-TiO_2_)/perovskite/MeO-PEAI/Spiro-OMeTAD/Au and further investigated the differences in photovoltaic performance based on PSCs with different ETLs. Figure 4a illustrates the reverse scan *J*–*V* curves of PSCs with champion PCEs based on TiO_2_ and PPA-TiO_2_. Impressively, PPA-TiO_2_ achieved a champion PCE of 24.83% with a short-circuit current density (*J*_SC_) of 25.43 mA/cm^2^, an open-circuit voltage (*V*_OC_) of 1.174 V, and a fill factor (FF) of 83.19%. Conversely, the PSCs with TiO_2_ reached a champion PCE of 24.21%, accompanied by a *J*_SC_ of 25.38 mA/cm^2^, a *V*_OC_ of 1.174 V, and an FF of 82.05%. The corresponding forward scan *J*–*V* curves of PSCs with different ETLs are shown in Appendix A. The results show that the *V*_OC_ and FF of the PSC based on the TiO_2_ ETL decrease significantly after the reverse scan, but the optimized PSC maintains a high *V*_OC_ and FF, and the reverse scan PCE is 24.68%. In order to reflect the hysteresis phenomenon of the PSCs more clearly, we calculated the hysteresis index (HI) using the formula of (PCE_reverse_ − PCE_forward_)/PCE_reverse_. The value of the HI for the TiO_2_-based PSC is 3.76%, while the value of the HI for the PPA-TiO_2_-based PSC is only 0.60%, which is almost negligible. The improvement in the hysteresis phenomenon is related to the charge extraction optimized at the ETL and perovskite interface, which reduces the accumulation of interface charges. The external quantum efficiency (EQE) spectra depicted in Figure 4b show that the integrated *J*_SC_ values of PSCs with TiO_2_ and PPA-TiO_2_ exhibit slight discrepancies compared to those derived from the *J*–*V* measurements. 

Figure 4c shows the statistical analysis results of the PCE values from 25 devices for each condition, and Appendix A shows the corresponding statistical distribution of *V*_OC_, *J*_SC_, and FF. The TiO_2_-based PSCs possess an average PCE of 23.45%. As a comparison, the PPA-TiO_2_-based PSCs possess a higher average PCE of 24.11%. The standard deviation (SD) is 0.705 for TiO_2_-based samples and 0.548 for PPA-TiO_2_-based samples, which indicates the higher repeatability of PPA-TiO_2_-based samples than TiO_2_-based samples. The increased PCE and reproducibility mainly result from the improved and reliable ETL quality after incorporating PPA. A more detailed analysis of the other performance parameters of PSCs shows that the *V*_OC_, *J*_SC_, and FF are all improved, which is conducive to the improvement of the average PCEs of the PSCs. It is worth noting that the increase in the FF is the most significant, from an average of 79.89% for TiO_2_-based PSCs to 81.54% for PPA-TiO_2_-based PSCs, which is related to the improvement of ETL quality and interface contact and defects after PPA optimization. In addition, we also characterized the device’s stability. Long-term storage tests on unencapsulated devices were performed in ambient conditions (30% RH, 25 °C). As shown in Figure 4d, we can observe that the unencapsulated device with PPA-TiO_2_ retained about 94% of its initial PCE after storage for 288 h, while the device with TiO_2_ retained about 89% of its initial PCE. Based on these results, it can be seen that PPA-TiO_2_ can effectively improve the PCEs of PSCs and also have a positive effect on device stability.

## 3. Conclusions 

In this work, we developed high-efficiency PSCs with an FF of more than 83% by regulating TiO_2_ deposition using PPA. Owing to the modified energy level structure and optimized interfacial contact property resulting from a smooth TiO_2_ surface, the PSCs with PPA-TiO_2_ possessed enhanced electron transport. Accordingly, the PSCs with PPA-TiO_2_ achieved a champion PCE of 24.83%, which was higher than that (24.21%) of PSCs with TiO_2_. In addition, the PSCs with PPA-TiO_2_ retained approximately 94% of their initial PCE after storage for 288 h in ambient conditions, showing enhanced storage stability. We believe that this deposition regulation strategy using single-anchored ligands can exhibit wide applicability using various functional materials (Appendix A) and could also be extended to regulate the deposition of other ETLs, which is significant for fabricating high-quality ETLs and directly understanding the influence of ETL modifications on the photovoltaic performance of PSCs, providing a reference for promoting the further development of planar PSCs.

## 4. Experimental Section

### 4.1. Materials

The FTO glass substrate (~13 ohm); TiCl_4_ (Aladdin, Wallingford, CT, USA); Phenylphosphonic acid, Isopropanol, and Acetonitrile (Acros, Waltham, MA, USA); Lead iodide, DMF, DMSO, and Chlorobenzene (Sigma-aldrich, St. Louis, MO, USA); FAI, MACl, CsI, 4-meo-PEAI, Spiro-OMeTAD, Lithiumbis (trifluoromethylsulfonyl) imide; and 4-tertbutylpyri-dine were from Xi’an Polymer Light Technology Corp, Xi’an, China. The deionized water was purified using the UPR-I ultrapure water machine developed by Upupure Technology Co., Ltd, Chengdu, China. We did not purify any chemicals further.

### 4.2. Device Fabrication

For the preparation of the FTO glass substrates, the FTO glass substrates were placed in custom cleaning boxes soaked in deionized water, ethanol, and deionized water. They were ultrasonically cleaned in an ultrasonic cleaning machine for 20 min each. Plasma cleaning was performed for 20 min after drying the surface moisture with a nitrogen gun, which helped to clean the surface of organic impurities and caused the formation of hydrophilic groups on the surface of the substrates.

### 4.3. ETL Preparation

A mixture of TiCl_4_ (4 mL) and deionized water (200 mL) was stirred to obtain a TiO_2_ precursor. As for the preparation of PAA-TiO_2_ ETL, phenylphosphonic acid was added to the TiO_2_ precursor solution at a ratio of 10 milligrams per milliliter of TiCl_4_. The FTO substrate was placed in a custom CBD container, poured into a pre-prepared precursor to completely soak the substrate, and transferred to a water bath heater set at 70 °C for 40 min to form the ETLs. After the preparation, the ETLs were removed from the chemical bath precursor, and the surface was washed with deionized water, ethanol, and deionized water successively to remove impurities and then dried with N_2_ for use. 

For the preparation of perovskite, the 1.54 M FA_0.85_MA_0.1_Cs_0.05_PbI_3_ (DMF: DMSO, 4:1 volume/volume) precursor was spin-coated on the FTO/ETLs substrate at 4000 rpm for 18 s and 800 μL ether was dropped after 6 s of rotation. At the end of the spin-coating process, the film was transferred to air (25 °C and 35% RH) and annealed on a hot table at 150 °C for 10 min. 

For the preparation of the 4-meo-PEAI layer, 4.5 mg/mL of 4-meo-PEAI dissolved in IPA was spin-coated on a perovskite surface at 4000 rpm for 30 s and annealed on a hot table at 100 °C for 3 min in N_2_.

For the preparation of the hole transport layer, a Spiro-OMeTAD solution was prepared by mixing 72.3 mg of Spiro-OMeTAD in 1 mL of chlorobenzene with 26.6 μL of TBP and 18 μL of Li-TFSI salt (520 mg mL^−1^ in ACN), which was spin-coated at 4000 rpm for 30 s in N_2_.

To obtain the PSCs, an Au electrode (60 nm) was deposited via thermal evaporation. After preparation, the PSCs were transferred to a drying cabinet (25 °C and 15% RH) for oxidation for 12 h.

### 4.4. Characterization

Cold field-emission scanning electron microscopy (SEM) using a Hitachi S-4800 instrument was used to analyze the surface morphology of the samples. X-ray photoelectron spectroscopy (XPS) and ultraviolet photoelectron spectroscopy (UPS) were performed using an ESCALAB 250Xi system to analyze the surface composition and electronic structure of the samples. Ultraviolet–visible (UV–Vis) spectra were recorded using a SHIMADZU UV-2600 spectrophotometer to study the optical properties of the films. Atomic force microscopy (AFM), Kelvin probe force microscopy (KPFM), and conducting atomic force microscopy (C-AFM) using an FMNanoview 1000 instrument were used to characterize the surface characteristics. Steady-state photoluminescence (PL) was employed to study the carrier behavior of the samples. The excitation wavelength used for these measurements was 470 nm. The Mott–Schottky curves of the solar cells were measured using an electrochemical workstation (Zahner Zennium), the disturbed AC voltage was 50 mV, the frequency was 10 KHz, the step width was 0.01 V, and the bias voltage range was from −0.5 to 1.3 V. The device efficiency was measured using a Keithley 2400 source meter with a scan rate of 13 mV s^−1^ under simulated AM 1.5G illumination (100 mW cm^−2^) provided by a 150 W Class AAA solar simulator (XES-40S1, SAN-EI). The external quantum efficiency and integrating current of the perovskite solar cells were measured using QE-R systems (Enli Tech, Shanghai, China) under AC pattern conditions. 

## Figures and Tables

**Figure 1 materials-17-03820-f001:**
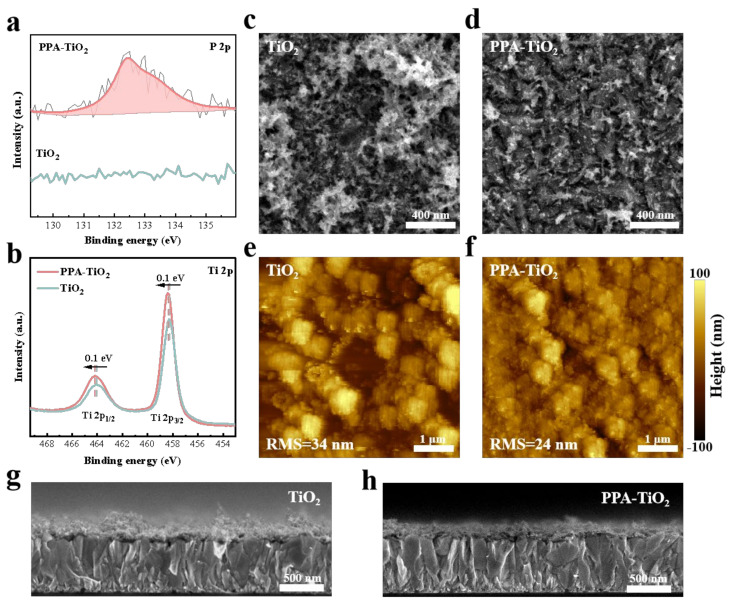
Characterization of the ETLs. (**a**) XPS spectra of P 2p for the ETLs. (**b**) XPS spectra of Ti 2p for the ETLs. (**c**,**d**) Surface SEM images of the ETLs. (**e**,**f**) AFM images of the ETLs. (**g**,**h**) Cross-sectional SEM images of the ETLs.

**Figure 2 materials-17-03820-f002:**
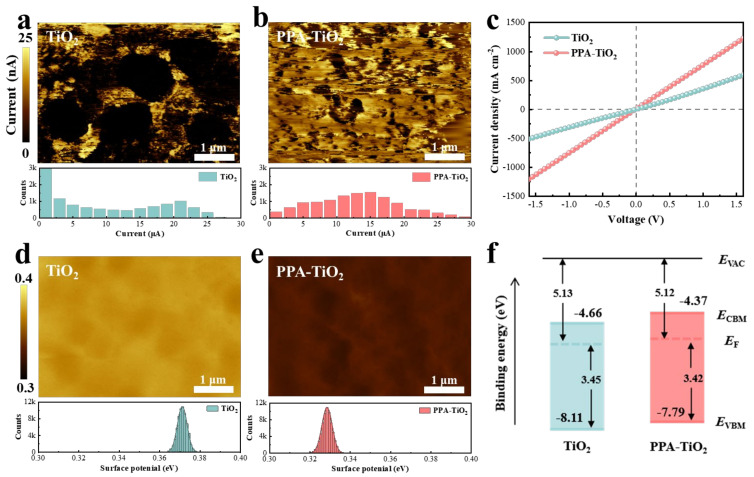
Characterization of the ETLs. (**a**,**b**) C-AFM images of the TiO_2_ and PPA-TiO_2_ films. The statistical current distributions of the film surfaces are shown at the bottom. (**c**) *J*–*V* curves of the devices structured as FTO/ETLs/Au. (**d**,**e**) Surface potential images of the TiO_2_ and PPA-TiO_2_ films. The statistical potential distributions of the film surfaces are shown at the bottom. (**f**) Energy level diagram of TiO_2_ and PPA-TiO_2_.

**Figure 3 materials-17-03820-f003:**
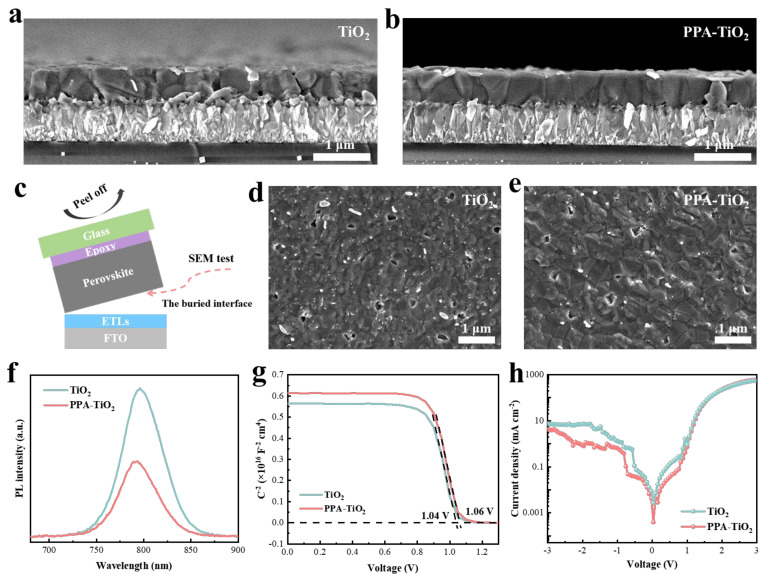
Physical properties of PSCs with different ETLs. (**a**,**b**) Cross-sectional SEM images of perovskite films deposited on different ETLs. (**c**) Schematic illustration of exposing the buried interface for SEM testing. (**d**) SEM images for perovskite at the buried interface based on the FTO/TiO_2_/perovskite films. (**e**) SEM images for perovskite at the buried interface based on the FTO/PPA-TiO_2_/perovskite films. (**f**) PL spectra of the perovskite films deposited on different ETLs. (**g**) Built-in electric field identified from the Mott–Schottky curves of devices based on different ETLs. (**h**) Dark *J*–*V* curves of the PSCs with TiO_2_ and PPA-TiO_2_.

**Figure 4 materials-17-03820-f004:**
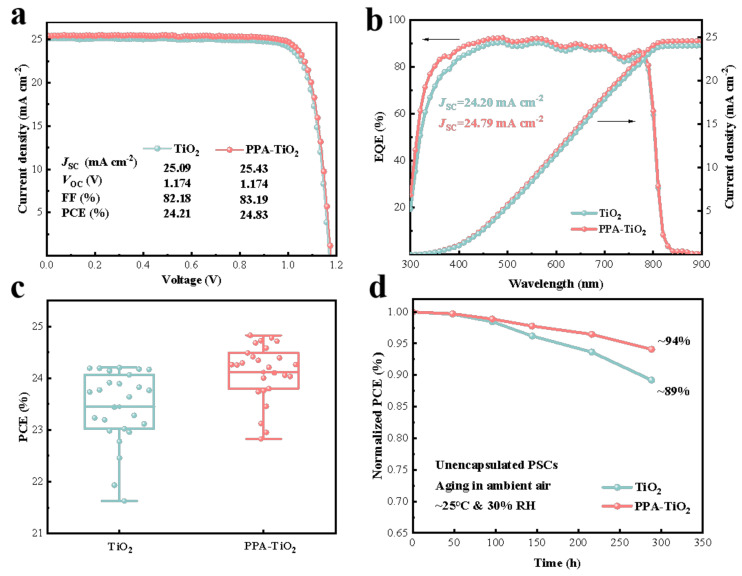
Photovoltaic performance of PSCs with various ETLs. (**a**) *J*–*V* curves (reverse scans) of the champion PSCs based on the TiO_2_ and PPA-TiO_2_ ETL. (**b**) EQE and integrated current density curves of PSCs based on the TiO_2_ and PPA-TiO_2_ ETL. (**c**) Distribution of the PCE values from 25 devices for each group. (**d**) PCE evolution of 25 unencapsulated devices stored in ambient conditions (30% RH, 25 °C).

## Data Availability

The original contributions presented in the study are included in the article/Appendix A, further inquiries can be directed to the corresponding author.

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
