# Peer review of "Regulating TiO2 Deposition Using a Single-Anchored Ligand for High-Efficiency Perovskite Solar Cells"

_materials, 2024, doi:10.3390/ma17153820_

Round 1

Reviewer 1 Report

Comments and Suggestions for Authors

In this short article, the Authors have shown have they have been able to regulate the TiO2 chemical bath deposition by incorporating the phenylphosphonic acid into the bath precursor that was mixed with TiCl4 and deionized water. The manuscript is concise, yet very informative, and the results are nicely presented. However, the work should also be revised, my detailed comments can be found below.

Line 32, what is this limit (in %)?

Figure S1 (structure) should be moved to the introduction, since it is being discussed in lines 63-43

Figure 4c, the authors describe it as a statistical analysis while this is just a data presentation. In fact, the Authors should have used the statistical analysis to check if there are statistically significant differences in PCE between TiO2 and PPA-TiO2

Line 254, it should be stated whether it was lead(II) or lead(IV) iodide

Materials, water. Sometimes the authors write that they have used “ultrapure” and sometimes “deionized” water. Was it really so? The type of water and method of its purification should be clearly stated in this section.

Since the application of PPA turned out to be very successful, the Authors should postulate (in conclusions) the further research direction. For example, the application of phenylphosphinic acid seems to be a nice choice for further evaluation.

Author Response

Reviewer #1:

In this short article, the Authors have shown have they have been able to regulate the TiO2 chemical bath deposition by incorporating the phenylphosphonic acid into the bath precursor that was mixed with TiCl4 and deionized water. The manuscript is concise, yet very informative, and the results are nicely presented. However, the work should also be revised, my detailed comments can be found below.

Response:

We appreciate your recognition and recommendation of our work. We have carefully revised our manuscript according to your insightful comments one by one. We sincerely express our admiration for the professional review which effectively helped our manuscript to be better.

Comments 1: Line 32, what is this limit (in %)?

Response:

The Shockley-Queisser limit here refers to the maximum efficiency that can be achieved based on theoretical calculations of a single-junction solar cell, and its value is 33.7%.

Comments 2: Figure S1 (structure) should be moved to the introduction, since it is being discussed in lines 63-43.

Response:

Thank you for your consideration and valuable comments. According to your suggestion, we have adjusted the structural formula of phenylphosphonic acid (Figure S1) to the introduction part.

The revised part in the main text on page 3: “Herein, we regulate the TiO2 CBD by incorporating the phenylphosphonic acid (PPA, the molecular structure formula shown in Figure S1) into the bath precursor that was mixed with TiCl4 and deionized water.”

Comments 3: Figure 4c, the authors describe it as a statistical analysis while this is just a data presentation. In fact, the Authors should have used the statistical analysis to check if there are statistically significant differences in PCE between TiO2 and PPA-TiO2.

Response:

Thank you for your consideration and useful comments. We only showed the distribution of efficiency in the original manuscript. According to your suggestion, we further made a statistical analysis of the distribution of PCE and calculated standard deviation (SD) to evaluate the repeatability of the sample. The result showed an SD of 0.705 for TiO2-based samples and 0.548 for PPA-TiO2-based samples, which indicates the higher repeatability of PPA-TiO2-based samples than TiO2-based samples. The increased PCE and reproducibility mainly result from the improved and reliable ETL quality after incorporating PPA.

The revised part in the main text on page 10: “The standard deviation (SD) of 0.705 for TiO2-based samples and 0.548 for PPA-TiO2-based samples, which indicates the higher repeatability of PPA-TiO2-based samples than TiO2-based samples. The increased PCE and reproducibility mainly result from the improved and reliable ETL quality after incorporating PPA.”

Comments 4: Line 254, it should be stated whether it was lead(II) or lead(IV) iodide.

Response:

Thank you for your meticulous question. The exact form of writing is Lead(II) iodide.

The revised part in the main text on page 7: “Lead(II) iodide”

Comments 5: Materials, water. Sometimes the authors write that they have used “ultrapure” and sometimes “deionized” water. Was it really so? The type of water and method of its purification should be clearly stated in this section.

Response:

Thank you for your consideration and valuable comments. We use deionized water in our experiments. According to your suggestion, we have unified the writing inconsistencies in the original manuscript. In addition, the deionized water be purified by the UPR-I ultrapure water machine developed by UPU-Pure Technology Co., Ltd.

The revised part in the main text on page 12: “The deionized water be purified by the UPR-I ultrapure water machine developed by Upupure Technology Co., Ltd.”

Comments 6: Since the application of PPA turned out to be very successful, the Authors should postulate (in conclusions) the further research direction. For example, the application of phenylphosphinic acid seems to be a nice choice for further evaluation.

Response:

We appreciate your insightful suggestion. As you said, the proposed deposition regulation strategy of single-anchored ligand also provides a reference for selecting other materials to use for ETL deposition regulation, such as phenylphosphinic acid, benzene sulfonic acid, dioctyl phenylphosphonate, etc, which provides more possibilities for the production of high-quality ETL and efficient PSCs.

The revised part in the main text on page 11: “We believe that this deposition regulation strategy using single-anchored ligands can guide the selection of similar functional materials and also be extended to regulate the deposition of other ETL.”

Reviewer 2 Report

Comments and Suggestions for Authors

Manuscript ID: materials-3060667

Title: Regulating TiO2 deposition using single-anchored ligand for

high-efficiency perovskite solar cells 

Authors: Zhanpeng Xu, Zhineng Lan, Fuxin Chen, Chong Yin, Longze Wang, Zhehan Li, Luyao Yan, Jun Ji * 

The manuscript reports on the fabrication of perovskite solar cell prototypes that exhibit improved power conversion efficiency and enhanced chemical stability under ambient conditions. This topic is relevant to the field. Specifically, the authors have functionalized the TiO2-based electron transport layer with phenylphosphonic acid. This functionalization appears to slightly improve the mentioned properties.

The enhancements, albeit present, are rather modest, leaving uncertainty regarding avenues for further improvement.

In the current version, it seems like the authors chose the agent for functionalization almost randomly, based on just two criteria: the need for a chemically active group capable of forming a X-O-Ti chemical bond, and the requirement for a bulky substituent at this group. This strategy doesn't look promising because it relies on luck rather than planning.

Specific improvements to consider.

Why should the inhibition of particle aggregation through steric hindrance result in improved power conversion efficiency and enhanced chemical stability?

Why has phenylphosphonic acid been specifically chosen as the functionalization agent?

Comments on the Quality of English Language

The English in the manuscript is readable but could benefit from grammar editing, especially regarding the use of articles.

Author Response

Reviewer #2:

The manuscript reports on the fabrication of perovskite solar cell prototypes that exhibit improved power conversion efficiency and enhanced chemical stability under ambient conditions. This topic is relevant to the field. Specifically, the authors have functionalized the TiO2-based electron transport layer with phenylphosphonic acid. This functionalization appears to slightly improve the mentioned properties.

The enhancements, albeit present, are rather modest, leaving uncertainty regarding avenues for further improvement.

In the current version, it seems like the authors chose the agent for functionalization almost randomly, based on just two criteria: the need for a chemically active group capable of forming a X-O-Ti chemical bond, and the requirement for a bulky substituent at this group. This strategy doesn't look promising because it relies on luck rather than planning.

Response:

We thank you for taking the time to review our manuscript and giving valuable comments. Based on your professional suggestions, we have carefully revised the manuscript and made appropriate revisions and supplements. We have replied to the questions point-by-point.

First, the single-anchored ligand regulation strategy proposed by us is a general solution to the aggregation problem in the CBD process of TiO2. On the one hand, coordination groups in the ligand molecules are used to achieve stable binding with TiO2, and on the other hand, groups with steric hindrance effects are used in the ligand molecules to inhibit aggregation during deposition (Figure R1). It can effectively improve the quality of the TiO2 ETL, optimize the interface contact and electron transport between the ETL and perovskite, and help to improve the performance of PSCs.

It's worth noting that the single-anchored ligand regulation strategy is universal. Theoretically, it only needs to find materials that meet the above two conditions to realize the adjustment of the CBD process of TiO2, such as phenylphosphinic acid, benzene sulfonic acid, dioctyl phenylphosphonate, β-Guanidinopropionic acid, etc. The phenylphosphonic acid additive we use in this paper is only a representative example because it contains a phosphonic acid group that can coordinate TiO2 and a sterically hindered benzene ring.

To demonstrate the universality of this strategy, we used β-Guanidinopropionic acid (Figure R2), which contains a carboxyl group that can bind to TiO2, and the molecular skeleton and the guanidine group that acts as the steric hindrance, to regulate the CBD process of TiO2. As predicted, many dendrite-like aggregates appear on the surface of the normal TiO2 ETL, which makes the surface very rough. However, the surface of the ETL film with β-Guanidinopropionic acid regulation is smoother, and there is no obvious agglomeration (Figure R3).

In short, the deposition regulation strategy using single-anchored ligands is proposed through rational and comprehensive thinking about the chemical deposition process and the ligand configuration. In this work, the phenylphosphonic acid is only a representative example to detailly demonstrate the proposed strategy. Except for the phenylphosphonic acid, the β-Guanidinopropionic acid that contains a binding group and a linear structure was also utilized to verify the effectiveness of this strategy. Our strategy can be performed through various materials, just including phenylphosphonic acid, which proves that this strategy exhibits wild applicability and excellent operability.

Figure R1. Schematic diagram of mechanisms regulating CBD processes of TiO2.

Figure R2. The molecular formula of the β-Guanidinopropionic acid.

Figure R3. Surface SEM images of ETLs, respectively.

The revised part in the main text on page 11: “We believe that this deposition regulation strategy using single-anchored ligands can exhibit wild applicability using various functional materials (Figures S7 and S8 and note S1) and also be extended to regulate the deposition of other ETL.”

The revised part in the Si on pages 9-10:

Figure S7. The molecular formula of the β-Guanidinopropionic acid.

Figure S8. Surface SEM images of ETLs, respectively.

Note S1.To demonstrate the universality of the single-anchored ligand regulation strategy, we used β-Guanidinopropionic acid (Figure S2), which contains a carboxyl group that can bind to TiO2, and the molecular skeleton and the guanidine group that acts as the steric hindrance, to regulate the CBD process of TiO2. As predicted, many dendrite-like aggregates appear on the surface of the normal TiO2 ETL, which makes the surface very rough. However, the surface of the ETL film with β-Guanidinopropionic acid regulation is smoother, and there is no obvious agglomeration (Figure S3).

Comments 1: Why should the inhibition of particle aggregation through steric hindrance result in improved power conversion efficiency and enhanced chemical stability?

Response:

Thank you for your wise question. In the CBD process of TiO2, the violent reaction of TiCl4 will lead to the formation and aggregation of a large number of TiO2 particles in a short time. The structure of the agglomerated TiO2 dendrites is not dense, resulting in poor conductive properties, and the uneven distribution leads to a rough surface of the ETL, resulting in voids at the bottom interface of the perovskite film. It affects charge transfer and extraction, resulting in poor device performance. More importantly, the charge accumulation and voids at the interface will accelerate the degradation of the interfacial material and intensify the ion migration within the film, thus affecting the stability of the PSCs. (ACS Energy Lett. 2020, 5, 2580−2589)

The introduction of materials with coordination ability and steric hindrance effect in the CBD process can slow down the deposition rate of TiO2 particles and effectively inhibit agglomeration to achieve the preparation of a dense and flat high-quality ETL. The flat and dense ETL helps to deposit the upper perovskite film and improve the contact between the ETL and the perovskite layer, which can effectively improve the interface charge extraction ability and lateral uniformity, and improve the performance of PSCs. It is worth noting that the effect of our strategy not only includes the steric hindrance effect but also can optimize the energy level structure between TiO2 and perovskite films, which will help to reduce interfacial charge accumulation and improve the interface and PSCs stability.

Comments 2: Why has phenylphosphonic acid been specifically chosen as the functionalization agent?

Response:

Thank you for your consideration and valuable comments. The phenylphosphonic acid additive we use in this paper is only a representative example because it can satisfy both requirements required by the single-anchored ligand strategy. It contains a phosphonic acid group that can coordinate TiO2 and a sterically hindered benzene ring. In addition, the incorporated phenylphosphonic acid can induce the upshift of the Fermi-level of TiO2 film, which is beneficial for interfacial electron transport.

The revised part in the main text on page 3: “For regulating CBD, we introduced PPA, as a representative example, into the chemical bath precursor in advance for the subsequent film growth.”

Comments 3: The English in the manuscript is readable but could benefit from grammar editing, especially regarding the use of articles.

Response:

Thanks for your careful concern and helpful comments. Based on your advisable comments, we have carefully checked and revised the inappropriate expression and further improved the expression of the manuscript.

Round 2

Reviewer 1 Report

Comments and Suggestions for Authors

The Authors have revised and improved their work. This version should be accepted.